# MAGF: A Statistically-Grounded Attention Mechanism for Multimodal Fusion in Early Alzheimer's Detection

**Abstract.** Early detection of Alzheimer's Disease (AD) is critical, yet diagnoses often rely on costly neuroimaging. We propose the Multi-View Attention-Guided Multimodal Fusion (MAGF) framework, a deep learning model for early AD detection using four accessible clinical data modalities: cognitive assessments, cerebrospinal fluid (CSF) biomarkers, genetic factors, and demographics. The core of our framework is a novel, parameter-free attention mechanism grounded in the coefficient of variation (CV), which posits that a modality's diagnostic utility is inversely proportional to its relative data dispersion. This yields an inherently interpretable fusion process where each modality's contribution directly reflects its statistical reliability. Evaluated on the ADNI cohort ($N = 1641$), MAGF achieves 89.7% accuracy and a 0.92 AUC, outperforming standard multi-head attention by 11.3% absolute accuracy. Our work introduces a transparent and statistically principled approach to multimodal fusion that addresses the need for trustworthy AI and mirrors the holistic reasoning of clinicians, paving the way for broader clinical deployment.

**Keywords:** Multimodal Fusion · Alzheimer's Disease · Interpretable AI · Attention Mechanism · Representation Learning · Clinical Decision Support

## 1 Introduction

Alzheimer's Disease (AD) is a progressive neurodegenerative disorder for which early detection is key, but conventional methods such as MRI or PET scans are often inaccessible. While this has motivated the use of more accessible clinical data, deep learning models in this domain frequently exhibit a "black-box" nature, limiting clinical trust and adoption. Rather than applying post-hoc explanations, we propose a new fusion mechanism whose interpretability is *intrinsic* and grounded in statistical theory from the outset. We challenge the paradigm of learned attention by introducing a parameter-free mechanism motivated by the hypothesis that a modality's diagnostic utility is inversely proportional to its relative dispersion (CV). This framework provides verifiable transparency, where a modality's weight is a direct and reliable computation rather than an opaque, learned parameter.

## 2   Proposed Method: The MAGF Framework

The core of MAGF is the CV-Attention module. To address the interpretability gap, we define each modality's reliability as the inverse of its coefficient of variation.

**Definition 1 (Modality Reliability).** For a modality embedding $m_i$, its reliability $R_i$ is the inverse of its batch-wise coefficient of variation:

$$R_i = \frac{|\mu_i|}{\sigma_i + \epsilon}, \tag{1}$$

where $\mu_i$ and $\sigma_i$ denote the batch-wise mean and standard deviation computed across feature dimensions of modality $i$. The corresponding attention weights $\alpha_i^*$ are directly proportional to reliability:

$$\alpha_i^* = \frac{R_i}{\sum_j R_j} = \frac{|\mu_i|/\sigma_i}{\sum_j (|\mu_j|/\sigma_j)}. \tag{2}$$

These weights are therefore interpretable as measures of relative statistical reliability within a batch. Computing CVs batch-wise allows the model to dynamically adjust modality weights based on local statistics, mimicking clinical reasoning.

**Implementation Details.** Each of the four modalities is processed by a specialised MLP encoder. All encoders use ReLU activations and a dropout rate of 0.3. The Cognitive encoder is a two-layer MLP ($Dense_{128} \rightarrow Dense_{64}$). The Biomarker encoder applies a $Dense_{64}$ layer after custom "RatioAugment" and scaling layers. The Genetic encoder combines a 16-dimensional APOE embedding with a $Dense_{32}$ layer for the polygenic risk score. The Demographic encoder processes categorical and numerical features via a $Dense_{64}$ layer. The final fused representation $f = \sum_{i=1}^{m} \tilde{\alpha}_i m_i$ uses a square-root damping function $\tilde{\alpha}_i = \sqrt{\alpha_i^*}/\sum_j \sqrt{\alpha_j^*}$ to prevent modality dominance. The model is trained end-to-end using weighted cross-entropy loss with L2 regularisation and Adam optimisation.

## 3   Experiments and Results

We evaluated MAGF on the ADNI cohort ($N = 1641$) using a stratified 5-fold cross-validation protocol, benchmarking against 10 baselines. As shown in Table 1, MAGF achieved a mean accuracy of 89.7% and an AUC of 0.92, an 11.3% absolute gain over standard multi-head attention (MHA).

Ablation studies (Table 2) validate the model design. Removing cognitive data caused the largest drop (49.85%), while replacing CV-attention with uniform weighting led to a substantial 34.1% accuracy decline, demonstrating its critical role.

A SHAP analysis confirmed the model's clinical relevance, identifying ADAS-Cog scores, the $A\beta_{1-42}$/T-tau ratio, and FAQ scores as the most influential predictors, aligning with established clinical knowledge.

Table 1: Comparative performance on the ADNI cohort. MAGF consistently outperforms all baselines.

| Method | Accuracy (%) | AUC | Sens. (%) | Spec. (%) | F1-Score |
|---|---|---|---|---|---|
| **MAGF (Proposed)** | **89.7 ± 2.3** | **0.92 ± 0.03** | **88.7 ± 2.8** | **91.2 ± 2.1** | **0.879 ± 0.027** |
| Standard MHA | 78.4 ± 3.1 | 0.79 ± 0.04 | 76.2 ± 3.5 | 80.1 ± 3.2 | 0.772 ± 0.031 |
| Hybrid LSTM-FNN | 87.3 ± 2.5 | 0.90 ± 0.03 | 85.1 ± 2.9 | 89.7 ± 2.6 | 0.867 ± 0.028 |
| MAFDSRP | 85.1 ± 2.7 | 0.88 ± 0.03 | 82.7 ± 3.2 | 87.4 ± 2.8 | 0.847 ± 0.029 |
| Late Fusion Ens. | 82.1 ± 2.8 | 0.84 ± 0.03 | 79.8 ± 3.1 | 84.7 ± 2.9 | 0.815 ± 0.028 |
| XGBoost | 75.2 ± 3.6 | 0.75 ± 0.04 | 71.8 ± 4.1 | 76.9 ± 3.7 | 0.745 ± 0.037 |

Table 2: Systematic ablation of model components. Results correspond to a representative split used for ablation; Table 1 reports cross-validated averages.

| Configuration | Accuracy (%) | Drop (%) | Cohen's d |
|---|---|---|---|
| Full MAGF | 93.01 ± 1.8 | 0.00 | – |
| No Attention (Uniform) | 58.91 ± 3.2 | 34.10 | 12.7 |
| No Cognitive | 43.16 ± 3.8 | 49.85 | 15.2 |
| No Biomarkers | 68.09 ± 3.1 | 24.92 | 8.9 |
| No Genetic | 87.45 ± 2.1 | 5.56 | 2.8 |

## 4    Discussion and Conclusion

This work presents a transparent approach to multimodal fusion by grounding attention in statistical reliability, addressing the need for interpretable and trustworthy AI in healthcare. The interpretable weights act as a "confidence report," providing clinicians insight into modality-level certainty.

A key implication is the potential for **adaptive experimental design**. Since CV-based weights quantify real-time confidence, the model could identify modalities with high uncertainty (high CV) for a given patient—informing when expensive biomarkers are needed versus when cognitive data suffice. This framework thus aligns with efficient, data-driven experimental prioritization.

MAGF delivers strong empirical performance while maintaining interpretability. The $1/CV$ relationship remains a heuristic under the assumption of approximately Gaussian embeddings, but it represents a first step toward statistically grounded fusion models. Future work will extend this framework to longitudinal and external-cohort validation.

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
