# OpenReview forum: "MAGF: A Statistically-Grounded Attention Mechanism for Multimodal Fusion in Early Alzheimer’s Detection"
_AJCAI/2025/Workshop/AIML-CEB — AIML-CEB 2025 Poster_

### Official Review · Reviewer_i4Sk · 2025-11-04
**Interesting idea, but further clarifications and justifications needed to confirm the validity and soundness of proposed method.**

**Rating:** 5
**Confidence:** 4

**Review:**

The paper proposes a new parameter-free formula for calculating the attention weights for multimodal predictions of AD. The attention weight of each modality is designed to be proportional to inverse batch-wise coefficient of variation, and a final fused representation is derived as the weighted sum of each modality's representation. The model was tested on the ADNI dataset and outperformed all competing methods.

Side note: The in-text citations weren't showing up in the submitted pdf, and the reviewer wasn't able to get the full context intended by the author(s)

Pros:

1. As far as the reviewer is aware, the proposed method is original and novel. The proposed attention mechanism is not QKV based learned attention, but a weight score assigned to each modality based on the data dispersion of its features, which in itself is learned through feedforward layers.  The method outperformed all baselines on the benchmark dataset, which suggests good algorithmic power.

2. The study deals with the topic of multimodal medical diagnostics, which is of significant relevance. The study is aware that some modalities hold more diagnostic power than others, which becomes the motivation of the design.

3. The CV-based attention weights is clearly defined in the paper.

Cons:

1. The model architecture, aside from the attention weights, is not sufficiently defined. It is not immediately clear what is included in the attention mechanism aside from the proposed attention weights. It is also unclear what happens after the final fused representation f is derived, though one could make an educated guess that it will be fed into a classification head. On that note, the classification task is also not clearly defined.
It is also unclear what information is available from each modality (especially genetic factors), or what is the exact data format of each modality. However, the reviewer is aware that the page limit might contribute to the problem.

2. The study is based on the hypothesis that
> "a modality’s diagnostic utility is inversely proportional to its relative data dispersion"

which is not intuitive or well known to the general deep learning / biology audience. Experiments have shown good results, but experimental details are unclear, and the model was only tested on one specific dataset. More evidence, either theoretical or experimental, are needed to support the method's general robustness.

3. One might argue that by making the attention weights statistically grounded, batch-dependant and not directly learnable, the model becomes less adaptive, not more. Further justifications or context are needed to explain why such modifications are beneficial to the scenario of interest.
> "These weights are therefore interpretable as measures of relative statistical reliability within a batch. Computing CVs batch-wise allows the model to dynamically adjust modality weights based on local statistics, mimicking clinical reasoning."

Here, the "local" neighbourhood, presumably the local mini-batch of samples, does not hold credible information of  "local statistics", because sample batching is usually random and unrelated to inter-samples relations, unless it has been done differently here. If so, the author(s) need to declare their method and provide justifications. It's also difficult to see the connections to clinical reasoning.

4. Further elaborations needed for the claim:
> "This framework provides verifiable transparency, where a modality’s weight is a direct and reliable computation rather than an opaque, learned parameter."

The model explanatory analysis used in this study was SHAP, which is a black-box-applicable algorithm that could also works for the baseline with "opaque, learned parameters", presumably the original attention mechanism. It is not a sufficient proof of the aforementioned inherent transparency or interpretability of proposed model.

Comment:

1.
> "Early detection of Alzheimer’s Disease (AD) is critical, yet diagnoses often rely on costly neuroimaging. We propose ... model for early AD detection using four accessible clinical data modalities: cognitive assessments, cerebrospinal fluid (CSF) biomarkers, genetic factors, and demographics."

While neuroimaging can be fairly expensive, genetic factors aren't usually cheap to obtain, and extraction of cerebrospinal fluid is invasive and not so frequently practiced in clinical settings. Further justifications are needed to support the claim that having all four modalities available is more accessible than neuroimaging alone.

---

### Official Review · Reviewer_7DTr · 2025-11-07
**A new attention mechanism for batch-based data fusion**

**Rating:** 7
**Confidence:** 5

**Review:**

This is really interesting work, and quite relevant to the workshop. I think this could lead to a lot of interesting discussions.

The data-fusion mechanism you propose is really interesting, and I think the current work and results are suitable for presentation at the workshop.

As future work, it could be good to explore a few more directions, for example:

- I think your MAGF could be the same, or very similar to, a specific instance of a batch normalisation layer with fixed weights ($\gamma_i$ = some function of $R$, $\beta_i$ = 0) -- followed by a sum of dimensions. If so, this would also go some way to explaining the performance improvements you are seeing. It would be good to work through this fully and carefully point out the differences to the general case: https://en.wikipedia.org/wiki/Batch_normalization
- One could also consider a sample-wise attention mechanism based on epistemic model uncertainty, or some other gating mechanism. I recommend reading about [mixtures-of-experts](https://en.wikipedia.org/wiki/Mixture_of_experts), and possibly some old ideas like ["Bayesian committee machine"](https://www.dbs.ifi.lmu.de/~tresp/papers/bcm6.pdf) for inspiration. Instead of just using a predictor of the expectation, $f(x) \approx \mathbb{E}[y | x]$, you could consider using a model that gives you a predictive density, e.g. $\mathcal{N}(y|\mu(x), \sigma^2(x))$, and then you could use per-sample estimates of $\mu(x)$ and $\sigma(x)$ for an attention mechanism. It is pretty easy to turn regular neural networks into approximate Bayesian predictors, e.g. see [Simple and Scalable Predictive Uncertainty
Estimation using Deep Ensembles](https://proceedings.neurips.cc/paper_files/paper/2017/file/9ef2ed4b7fd2c810847ffa5fa85bce38-Paper.pdf).

---

### Official Review · Reviewer_2Bqy · 2025-11-09
**Optimizing attention mechanism for early alzheimer's detection**

**Rating:** 8
**Confidence:** 4

**Review:**

This paper proposes MAGF, a multimodal framework for early Alzheimer’s detection that fuses four non-imaging clinical modalities—cognitive tests, CSF biomarkers, genetics, and demographics—using a parameter-free attention derived from the coefficient of variation (CV) to yield interpretable modality weights.


Pros:

•Attempts to improve statistical interpretability with a parameter-free, CV-based attention.
•Good ablation study at the modality level.

Cons:

•It would be great if the authors could further analyze the batch-wise statistics, as they could introduce noise and instability during training.

•The implementation details of other models are not clearly described; variations in architectural choices (e.g., number of layers, embedding sizes in MHA) can substantially affect performance.

---

### Decision · Program_Chairs · 2025-11-12

Accept (Poster)